# Influence of Maternal Region of Birth on Placental Pathology of Babies Born Small

**DOI:** 10.3390/children9030388

**Published:** 2022-03-10

**Authors:** Mindi Fernando, Nalin Choudhary, Beena Kumar, Natasha Juchkov, Kathryn Shearer, Stacey J. Ellery, Miranda Davies-Tuck, Atul Malhotra

**Affiliations:** 1Monash Newborn, Monash Children’s Hospital, Melbourne, VIC 3168, Australia; mn.fernando@hotmail.com; 2Department of Paediatrics, Monash University, Melbourne, VIC 3168, Australia; nalinchoudhary94@gmail.com; 3Monash Health Pathology, Melbourne, VIC 3168, Australia; beena.kumar@monashhealth.org; 4Department of Obstetrics and Gynaecology, Monash University, Melbourne, VIC 3168, Australia; natasha.juchkov@gmail.com (N.J.); shearer.kaw@gmail.com (K.S.); stacey.ellery@monash.edu (S.J.E.); miranda.davies@monash.edu (M.D.-T.); 5The Ritchie Centre, Hudson Institute of Medical Research, Melbourne, VIC 3168, Australia

**Keywords:** neonate, pregnancy, small for gestational age

## Abstract

Background: Placental pathology is a common antecedent factor in infants born small for gestational age. Maternal region of birth can influence rates of SGA. Aims: To determine the association of maternal region of birth on placental pathology in babies that are born small, comparing a South Asian born population with Australia and New Zealand born women. Materials and methods: A retrospective cohort study was conducted at Monash Health, the largest public health service in Victoria. Mother-baby pairs above 34 weeks’ gestation and birth weight less than 10th centile born in 2016 were included. Placental pathology reports and medical records were reviewed. Statistical analyses of placental and selected neonatal outcomes data were performed. Results: Three hundred and eleven small for gestational age babies were included in this study, of which 171 were born to South Asian mothers and 140 to Australian and New Zealand mothers. There were no significant differences in gestational age at birth between the groups (38.7 (1.6) vs. 38.3 (1.7) weeks, *p* = 0.06). Placental pathology (macroscopic and microscopic) data comparisons showed no significant differences between the two groups (81% major abnormality in both groups). This was despite South Asian small for gestational age babies being less likely to require admission to a special care nursery or neonatal intensive care unit (35 vs. 41%, *p* = 0.05), or have a major congenital abnormality (2.3 vs. 4.3%, *p* = 0.04). Conclusion: In this observational study, maternal region of birth did not have an influence on placental pathology of babies born small, despite some differences in neonatal outcomes.

## 1. Introduction

Small for gestational age (SGA) defines infants born below the 10th percentile in birth weight for their gestational age [1]. This can be due to an inadequate supply of oxygen and nutrients during pregnancy and can include fetal growth restriction (FGR) infants [2]. Various maternal characteristics, including low body weight, short stature, nulliparity and maternal ethnicity or region of birth (ROB) are risk factors for SGA births [1]. With regards to maternal background, South Asian (SA) women have a higher risk of giving birth to an SGA infant compared with locally born women in high income countries [3]. In Australia, this is in comparison to those born to mothers of Australian or New Zealand (ANZ) ROB [3]. Difficulty remains in differentiating if higher rates of SGA babies amongst SA mothers are due to physiological (constitutional small and healthy) or pathological (growth restricted and associated with adverse outcomes) phenomena [4].

SGA and FGR are important pointers to fetal growth and are associated with perinatal morbidity and mortality. Perinatal and neonatal complications commonly seen in SGA and FGR infants include perinatal asphyxia, hypoglycaemia, polycythaemia, hypothermia, respiratory distress and stillbirth [5]. SGA babies born to women of SA ROB have different neonatal outcomes in comparison to those born to women of ANZ ROB [6]. In a recent study, it was found that SA SGA babies were at a higher risk of hypothermia and perinatal mortality compared to ANZ SGA babies; however, they had a lower risk of having congenital abnormalities, requiring respiratory support or nasogastric feeding [6]. It was suggested that placental structure and function might be a contributing factor to differing neonatal outcomes for these babies [6].

Placental insufficiency is the most common cause of growth restriction, but a number of fetal (chromosomal, congenital infection, metabolic disorders) and maternal (co-morbidities, malnutrition, lifestyle factors) factors can interplay to reduce growth [7,8,9]. Maternal region of birth may have an influence on placental structure and function as shown in some studies where SA women had higher rates of utero-placental vascular insufficiency as depicted by low placental weight with features of vasculopathy [4,10]. However, there is scarcity of literature evaluating placental pathology findings, specifically in an SGA population. We hypothesised that maternal ROB may affect placental pathology differentially in SGA babies.

This was a hypothesis generating study where we looked at SGA infants born over a one year period in a large health network. The primary aim of this study was to identify any distinguishing macroscopic or microscopic placental pathology characteristics between SGA infants born to SA or ANZ maternal ROB groups. The secondary aim was to assess selected neonatal outcomes between the groups.

## 2. Methods

A retrospective cohort study was conducted at Monash Health, the largest public health and maternity service in Victoria, Australia, providing care to over 9000 women per year across three hospitals. Monash Children’s Hospital, Clayton, provides paediatric healthcare services and is part of the Monash Health Network, and includes Monash Newborn, a tertiary neonatal intensive care unit (NICU) that provides care to around 1500 babies per year.

The study population included mothers of both SA and ANZ ROB who received their antenatal care and birthed at Monash Health sites between January and December 2016. Women were classified according to their ROB as defined by the United Nations geographical classification [11]. Women were identified as of South Asian origin if they were born in India, Pakistan, Bangladesh, Sri Lanka or Afghanistan. Women were identified as Australian or New Zealand origin if they were born in either country.

For these maternal groups, all maternal and birthing data for babies born ≥34 weeks’ gestation and under the 10th centile of birth weight for gestational age were extracted. Birth weight centiles were categorised in accordance with Australian national birth weight charts [12]. Exclusion criteria included pregnancies with multiple births, women who did not receive antenatal care at Monash Health and women born outside the SA and ANZ ROB. Further, those entries with incomplete placental pathology data were also excluded.

Databases utilised to collect data included the hospital’s Birthing Outcomes System (BOS) and Pathology results database. Data items extracted from BOS included maternal country of birth, age, parity, pre-existing medical co-morbidities and smoking, alcohol or substance use in pregnancy. Birth details extracted included gestation, type of birth, sex, weight, length, head circumference and Apgar scores. Selected neonatal outcomes assessed included whether the baby required respiratory support at birth, had congenital abnormalities or required admission to special care nursery (SCN) or NICU.

At Monash Health, birth weight under the 10th centile is recommended as an indication for performing placental histopathology to help identify underlying causes and pathogenesis of SGA. A detailed macroscopic and microscopic examination is carried out by a pathologist to include histological examination of membranes, umbilical cord, cord insertion site as well as fetal and maternal placental surfaces. Placental pathology results were obtained by searching each database using a hospital specific unique reference number for each patient. Placental histology results were extracted from this database and further categorised into macroscopic and microscopic findings. Macroscopic findings included trimmed placental weight, site of umbilical cord insertion, umbilical cord length, umbilical cord coiling or other cord abnormalities, number of vessels (2 vessels vs. 3 vessels) in the cord membrane completeness, maternal and fetal surface abnormalities and parenchymal abnormalities. Microscopic findings were grouped according to Amsterdam criteria and included maternal vascular malperfusion, fetal vascular malperfusion, delayed villous maturation, patterns of ascending intrauterine infection and villitis of unknown etiology [13].

Data analyses were performed using SPSS software (IBM version 23, Armonk, NY, USA). Macroscopic and microscopic placental pathology findings, as well as differences in maternal demographics and birth characteristics between SA and ANZ born women, were determined using t-tests, fisher’s exact test or chi-squared test depending on type of variable (continuous and categorical) and normality of data distribution. Normality was confirmed using a Shapiro—Wilk distribution test. Adjusted (for maternal age, BMI and smoking) *p* values for placental outcomes were also calculated. A *p* value of less than 0.05 was considered significant.

### Ethics Approval

This study was approved by the Monash Health Human Research Ethics Committee (HREC LNR/17/MonH/189RES-17-0000-216L).

## 3. Results

In 2016, 8525 singleton babies were born across Monash Health sites, of which 847 (9.9%) were SGA babies. Of these SGA births, only 432 (53.1%) babies received a placental examination and had a pathology report available. After exclusion criteria were applied, there were 311 SGA babies included in this study, of which 171 were born to SA ROB mothers and 140 were to ANZ ROB mothers (Figure 1).

### 3.1. Demographic Information

Demographics of the two groups are presented in Table 1. Women from SA ROB were older than those from ANZ ROB (31.1 vs. 29.0 years, *p* = 0.003) and had a lower body mass index (24.2 vs. 25.7, *p* = 0.01). SA born women had lower rates of smoking (1 vs. 46%, *p* = 0.0001) and alcohol consumption (1 vs. 4%, *p* = 0.04); however, they had higher rates of thyroid disease (11 vs. 6%, *p* = 0.04), gestational diabetes (9 vs. 1%, *p* = 0.004) and vitamin D deficiency (40 vs. 31%, *p* = 0.05). There were no significant differences between the two groups for pre-existing diabetes mellitus, hypertension, anaemia or polycystic ovarian disease. There was also no significant difference in family history of diabetes mellitus or hypertension; however, the incidence rate was high for both groups (SA 65 vs. ANZ 66%, *p* = 0.41).

### 3.2. Placental Outcomes

The associations between maternal ROB and placental pathology, both macroscopic and microscopic, are presented in Table 2, and representative photomicrographs of common placental pathology seen are shown in Figure 2. Placental weights were similar in the two groups (SA vs. ANZ; 376 vs. 383 g, *p* = 0.45). There was also no significant difference in umbilical cord length (329 vs. 320 mm, *p* = 0.43) between the two groups. Placental pathology was present in 81% of the placentae of women in both SA and ANZ groups. Common macroscopic findings included incomplete membranes (81 vs. 76%), and cord abnormalities were seen in 26.3% of SA SGA babies compared with 18.6% in ANZ cohort (*p* = 0.06). The most common placental microscopic pathologies identified according to the Amsterdam criteria included maternal vascular malperfusion (SA vs. ANZ; 81.3 vs. 86.4%, *p* = 0.28), fetal vascular malperfusion (66.7 vs. 57.1%, *p* = 0.09) and ascending uterine infection (43 vs. 39%, *p* = 0.60). Overall, there was no significant difference in macroscopic or microscopic placental pathology found between the two groups. This was true even after correcting for significant maternal demographic differences between groups (data not shown).

### 3.3. Selected Neonatal Outcomes

The differences between maternal ROB and selected neonatal outcomes are represented in Table 3. The distribution of SGA babies according to gestation for both groups is shown in Figure 3. Of those babies that had placental pathology reports completed, SA babies were born at a similar gestational age to ANZ babies (38.7 vs. 38.3 weeks, *p* = 0.05); however, they had a higher birth weight (2582 vs. 2497 g, *p* = 0.02), were less likely to require admission to SCN/NICU (35 vs. 41%, *p* = 0.05) or have a major congenital abnormality (2.3 vs. 4.3%, *p* = 0.04).

## 4. Discussion

This cohort study assessed the placental factors that may contribute to SGA babies born to two different maternal region of birth groups in an Australian setting. The study showed no significant differences in placental macroscopic and microscopic findings between SGA babies born to SA ROB women compared to ANZ ROB women, despite some differences in neonatal outcomes. The importance of this study is that it is the first of its kind in an Australian population, where influence of region of birth on placental pathology has been evaluated in SGA babies.

South Asian ROB is associated with an increased risk of SGA births [6,14,15]. In this study, there was a significant difference in demographic characteristics of SA and ANZ populations, with increased rates of alcohol and substance use identified in ANZ born women. In addition, the rate of reported smoking in the ANZ population was significantly higher at 46% compared with 1% in the SA born population. On the other hand, higher rates of thyroid disorders and gestational diabetes were reported for SA born women. These findings are in keeping with studies that have evaluated the relationship with perinatal smoking, alcohol and substance use as well as thyroid disease attributing to SGA [6,16,17,18,19]. Studies in multi-ethnic migrant and non-migrant populations have also concluded that biological and social factors including maternal age and smoking can have an impact on SGA births [20,21].

This study’s primary outcome was to look at placental pathology in the SA and ANZ populations to determine if ROB impacted on placental outcomes for SGA babies. Results showed four out of five SA SGA babies had significant pathology identified on placental pathology reports, which was similar to the ANZ born cohort. Placental weights and umbilical cord lengths between the two groups were similar. Placental findings showed a similar amount of maternal or fetal vascular malperfusion abnormalities. Ascending uterine infection was also seen in similar numbers (around 40%) in both groups. The single most common type of placental pathology identified was maternal vascular underperfusion, confirming that placental insufficiency is the most common type of pathology seen in SGA infants. Previous studies looking at placental factors that may attribute to SGA babies have found that SGA births have a direct correlation with smaller placentas, acute chorioamnionitis, higher rates of infarction, chronic villitis, fibrinoids and fetal vasculitis when compared with an appropriate gestational age cohort [22]. A study by Salafia et al. assessed placental lesions with their results showing that factors such as placental infarctions, hemorrhagic endovasculitis, chronic villitis and placental vascular thrombosis were significantly associated with growth restriction [23]. This is further supported by studies from the Indian subcontinent that have shown a higher incidence of low birth weight in babies born to women having placental insufficiency, with these studies concluding that placental dysfunction was more likely to explain low birth weight in comparison to maternal ethnicity or region of birth [24,25]. Our results were in support of these conclusions.

We also looked at placental literature in relation to small babies in other ethnic populations. Similar to SA ROB being known to have higher perinatal and neonatal morbidity, African-American women have increased rates of perinatal complications compared to White-American women [26]. Studies looking at maternal race and type of placental pathology have shown that African-American women had a higher risk of low placental weight as well as higher risk of fetal neutrophilic infiltration and decidual vasculopathy; however, the risk of placental vascular lesions such as thrombosis or infarcts, villous infarcts or fibrosis were lower in African-American women compared to White-American women [26,27]. This trend was consistent across placental findings for SGA babies in this population [26,27]. The increased risk of low placental weight and decidual vasculopathy in African-American women was thought to be due to an underlying vascular phenotype which made them more susceptible; however, this requires further investigation [27]. Placental findings in our study showed similar pathological processes to the above-mentioned studies such as placental vascular infarctions and villous infarcts attributing to growth restriction. The incidence of types of placental pathology was similar between SA and ANZ groups in our study though, in contrast to the findings seen in an American population.

South Asian ROB is thought to be associated with shortened gestational length with this resulting in more favourable outcomes for these neonates [6,21]. There is a suggestion that this could be due to earlier placental maturation at a biological level or due to genetic differences in SA women [6,16]. Specifically, higher cortisol levels in late gestation occur earlier in SA women leading to earlier lung maturation in these neonates, resulting in better outcomes [19]. Neonatal outcomes from this study showed SA babies had lower rates of congenital abnormalities and were less likely to need respiratory support at birth or require admission to SCN or NICU at birth compared to the ANZ cohort.

The neonatal outcomes we observed in this study are in keeping with previous literature, which report higher rates of congenital abnormalities or malformation syndromes in SGA babies from an ANZ background as well as having a higher incidence of requiring nasogastric feeding and longer length of stay during admission, when compared to SA babies [6]. This could suggest that maternal ROB has a differential impact on fetal growth, leading to the possibility of pathologically small infants in the ANZ population compared with a constitutionally small infant in the SA population [6,16]. The placental pathology findings of this current study suggest a similar pathological process across both groups, further supported by embryological studies concluding that fibrin deposition and dysregulated villous vasculogenesis are predominant findings in placentas of small babies if optimal placental development or compensatory mechanisms are impaired [9].

This observational study is the first of its kind in an Australian setting to assess the impact of maternal ROB on placental pathology. The strength of this study is that placental outcomes were looked at in detail. As data were collected over a 12-month period, sample size and data were limited. Only half of SGA babies in this time period had a completed placental pathology report and were able to be included in this study, limiting our ability to fully assess the role of placental pathology in SGA babies. A further limitation of this study was that appropriate for gestational age (AGA) babies were not included as a control group; however, the placental differences between AGA and SGA babies are well established [22]. Further, babies born under 34 weeks’ gestation were excluded from this study as that cohort is a small sample size and have established and varied risk factors for growth restriction (earlier onset, for example) as well as additional prematurity related complications. Placental inflammation commonly caused by infection is a known precipitating factor for prematurity, which would result in a higher incidence of acute chorioamnionitis in that cohort [28].

As no significant differences in placental pathology were observed in this study, future studies with a larger sample size and possibly broader inclusion criteria may be beneficial. On the other hand, South Asian ROB is diverse ethnically with women born in some South Asian countries being genetically and biologically different to women born in other South Asian countries [6]. Further, paternal ethnicity/ROB was not taken into account to determine whether paternal genetic factors could play a role. Future work accounting for some of these additional factors would be useful.

In this retrospective cohort study, placental pathology did not differ between infants born small to ANZ or SA maternal ROB, nor did it contribute to variations in neonatal outcomes between the cohorts, suggesting other factors are causative, but larger studies are required to confirm these findings.

## Figures and Tables

**Figure 1 children-09-00388-f001:**
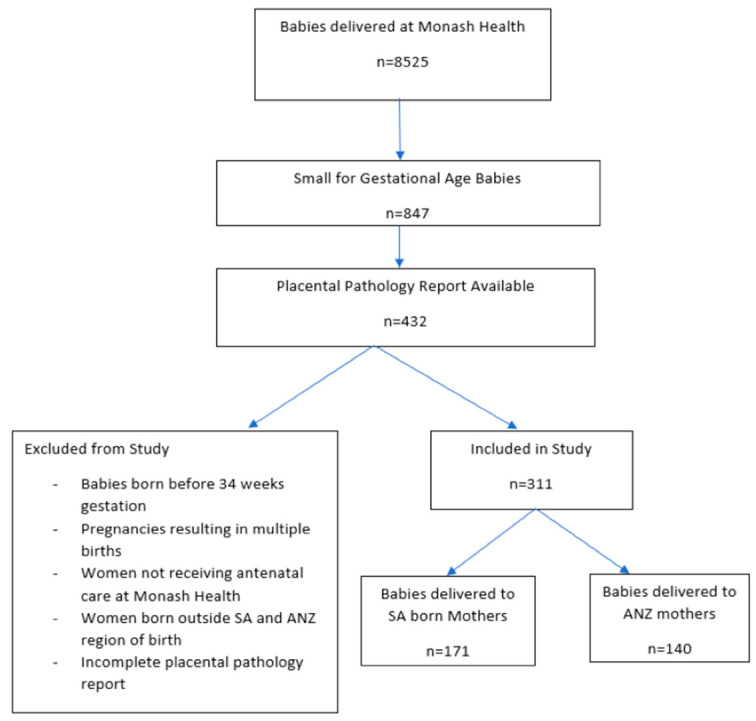
Flow diagram of included neonates.

**Figure 2 children-09-00388-f002:**
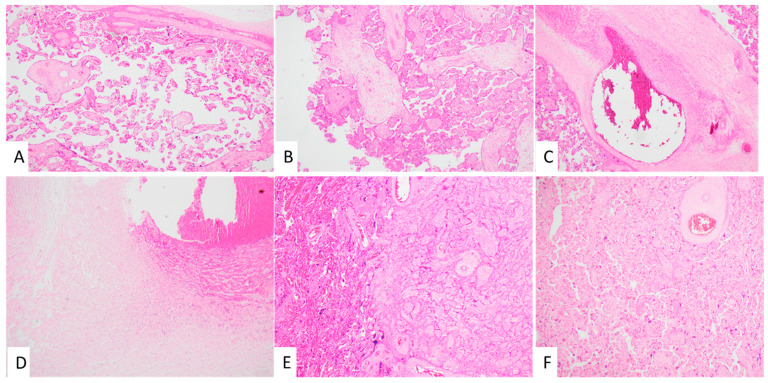
Representative photomicrographs (low magnification) of placental pathology seen in SGA infants. (**A**) Accelerated maturation, (**B**) chronic villitis, (**C**) fetal thromobotic vasculopathy, (**D**) funisitis, (**E**) placental infarct and (**F**) villous crowding, perivillous fibrin.

**Figure 3 children-09-00388-f003:**
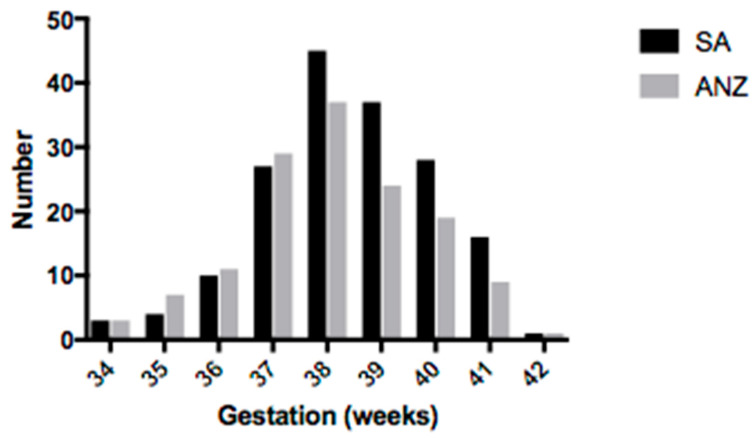
Distribution of small for gestational age babies according to gestational age.

**Table 1 children-09-00388-t001:** Maternal demographics.

	SA Infants (*n* = 171)	ANZ Infants (*n* = 140)	*p* Value
Maternal age (years)	31.16 (4.1)	29.04 (6.1)	0.0003
Body mass index (BMI)	24.27 (4.1)	25.73 (6.6)	0.01
Alcohol use in pregnancy	2 (1)	7 (4)	0.04
Smoking	1 (1)	64 (46)	0.0001
Maternal thyroid disease	18 (11)	9 (6)	0.04
Pre-existing hypertension	3 (2)	2 (1)	0.90
Gestational diabetes—insulin dependent	16 (9)	2 (1)	0.004
Maternal vitamin D deficiency	69 (40)	44 (31)	0.05
Nulliparous	88 (51)	79 (56)	0.78
Maternal diabetes mellitus	4 (2)	3 (2)	0.93
Maternal polycystic ovarian disease	9 (5)	3 (2)	0.22
Maternal anaemia	13 (8)	11 (8)	0.92
Family history of diabetes mellitus/hypertension	100 (65)	82 (66)	0.41
Spontaneous preterm labour (34–36 weeks)	17 (10)	21 (15)	0.26
Prolonged pregnancy > 41 weeks	14 (8)	10 (7)	0.71
Suspected FGR	70 (41)	58 (41)	0.56
Suspected fetal compromise	40 (23)	32 (23)	0.64
Pre-eclampsia	6 (3)	6 (4)	1.00
Antepartum haemorrhage	6 (4)	5 (4)	0.95

Data expressed as number (%) or mean (SD); missing information on some patients.

**Table 2 children-09-00388-t002:** Placental findings.

	SA Infants (*n* = 171)	ANZ Infants (*n* = 140)	*p* Value
Placenta weight, gms	376 (72)	383 (77)	0.45
Umbilical cord length, mm	329 (105.7)	320 (108)	0.43
Membranes complete	20 (19.4)	22 (23.7)	0.47
Maternal surface abnormalities	36 (21)	35(25)	0.41
- Thrombi	19 (11)	15 (11)	
- Calcification	1 (1)	1 (1)	
- Fibrin	1 (1)	3 (2)	
- Indentations	0 (0)	5 (4)	
Maternal vascular malperfusion	139 (81.3)	121 (86.4)	0.28
- Maternal vascular underperfusion features	62 (36)	52 (38)	
- Villous infarcts	28 (18)	29 (22)	
- Increased perivillous fibrin	38 (24)	28 (22)	
- Accelerated villous maturation	5 (3)	3 (2)	
- Intervillous fibrin	4 (2)	4 (3)	
- Villous agglutination	2 (1)	3 (2)	
- Maternal floor infarction	0 (0)	0 (0)	
- Haemorrhage	0 (0)	0 (0)	
Fetal vascular malperfusion	114 (66.7)	80 (57.1)	0.09
- Cord abnormalities	45 (26.3)	26 (18.6)	
- Signs of fetal vascular obstruction	6 (4)	10 (8)	
- Calcifications	5 (3)	6 (4)	
- Intervillous thrombi	27 (16)	24 (17)	
- Intervillous fibrin deposition	11 (6)	4 (3)	
- Meconium staining	20 (12)	10 (7)	
Delayed villous maturation	2 (1)	3 (2)	0.66
Ascending uterine infection	73 (43)	55 (39)	0.60
- Funisitis	5 (3)	5 (4)	
- Acute chorioamnionitis	38 (22)	30 (21)	
- Vasculitis	15 (9)	10 (7)	
- Acute Inflammation	15 (9)	10 (7)	
- Chronic inflammation	0 (0)	0 (0)	
Villitis of unknown etiology	26 (15)	15 (11)	0.83
Significant pathology identified	138 (81)	113 (81)	1.00

Data expressed as number (%) or mean (SD); missing information on some patients

**Table 3 children-09-00388-t003:** Selected neonatal outcomes.

	SA Infants (*n* = 171)	ANZ Infants (*n* = 140)	*p* Value
Birth weight, gms	2582 (316)	2497 (351)	**0.02**
Gestation, wks	38.7 (1.6)	38.3 (1.7)	0.05
Placental weight, gms	376 (72)	383 (77)	0.45
Feto-placental ratio	7.0 (1.1)	6.6 (1.3)	0.13
SCN/NICU admission	60 (35)	57 (41)	0.05
Respiratory support	28 (16.4)	25 (18)	1.0
Congenital abnormality	4 (2.3)	6 (4.3)	**0.04**

Data expressed as number (%) or mean (SD); statistically significant *p* values are shown in bold.

## Data Availability

Data available on reasonable request.

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
