# Peer review of "Influence of Maternal Region of Birth on Placental Pathology of Babies Born Small"

_children, 2022, doi:10.3390/children9030388_

Round 1
Reviewer 1 Report
An interesting article looking at the relationship between maternal region of birth with SGA.
Some suggestions for improvement:
Authors – Suggest to include a pathologist from the institution to oversee the placental pathology reports and diagnosis.
Line 101 - Placental weight – is the placenta being weighed together with the cord and membrane or is it a trimmed weight ?
Line 103 – Suggest to remove “microscopic findings included …” which should be in result section.
Line 119 – There were 847 SGA babies born in the year 2016 across Monash Health. Surprisingly, only about half of those babies have had their placenta being examined histologically. What are the reasons for that? Isn’t SGA an important indication for placental examination in accordance to College of American Pathologists (CAP) guideline?
Table 1 – how maternal vitamin D status being determined? Is it a routine investigation for all pregnancy complicated with SGA
Table 2 Placenta Findings – suggest to classify placental lesions in accordance to Amsterdam diagnostic criteria for placental reporting i.e. to categorize the findings into c/w maternal vascular malperfusion, fetal vascular malperfusion, inflammation etc.
Table 3 – suggest to add in fetoplacental weight ratio
Suggest to add in figures showing various pathological lesions seen in placentas that are associated with SGA in the present study.
Discussion - to discuss the placental pathological findings seen in SGA in both populations in accordance to Amsterdam diagnostic categories for placental reporting
Author Response
An interesting article looking at the relationship between maternal region of birth with SGA.
Some suggestions for improvement:
Authors – Suggest to include a pathologist from the institution to oversee the placental pathology reports and diagnosis.
Thank you for the suggestion. We have included Prof Beena Kumar from our institution.
Line 101 - Placental weight – is the placenta being weighed together with the cord and membrane or is it a trimmed weight ?
Placental weight is trimmed placental weight that is after removing the cord and trimming the membranes.
Line 103 – Suggest to remove “microscopic findings included …” which should be in result section.
Line 119 – There were 847 SGA babies born in the year 2016 across Monash Health. Surprisingly, only about half of those babies have had their placenta being examined histologically. What are the reasons for that? Isn’t SGA an important indication for placental examination in accordance to College of American Pathologists (CAP) guideline?
Yes, it is in our hospital guideline as well, but unfortunately not all placenta are sent for examination.
Table 1 – how maternal vitamin D status being determined? Is it a routine investigation for all pregnancy complicated with SGA
Vitamin D status is assessed routinely in our institution.
Table 2 Placenta Findings – suggest to classify placental lesions in accordance to Amsterdam diagnostic criteria for placental reporting i.e. to categorize the findings into c/w maternal vascular malperfusion, fetal vascular malperfusion, inflammation etc.
We have classified the placental lesions in accordance with the Amsterdam criteria and modified Table 2
Table 3 – suggest to add in fetoplacental weight ratio
Thank you for suggestion. We have added it to Table 3.
Suggest to add in figures showing various pathological lesions seen in placentas that are associated with SGA in the present study.
We have added representative photomicrographs as suggested.
Discussion - to discuss the placental pathological findings seen in SGA in both populations in accordance to Amsterdam diagnostic categories for placental reporting
Edited the discussion as suggested.
Reviewer 2 Report
This is a very interesting work, nicely and clearly presented.
The manuscript is overall fine and very interesting. There are very few minor spelling mistakes throughout the manuscript that I did not take specific note.
Author Response
This is a very interesting work, nicely and clearly presented.
The manuscript is overall fine and very interesting. There are very few minor spelling mistakes throughout the manuscript that I did not take specific note.
Thank you for your encouraging comments. We have done a very thorough proof read of manuscript as suggested.
Round 2
Reviewer 1 Report
Overall adequate corrections made.
Only some minor comments on the photos (figure 2).
The picture of chronic villitis, fetal thrombotic vasculopathy and funisitis are not clear. A clearer picture to show the exact histo findings is recommended.